# Landfill Levy Imposition on Construction and Demolition Waste: Australian Stakeholders' Perceptions

**Salman Shooshtarian \***, **Tayyab Maqsood, Malik Khalfan, Rebecca J. Yang and Peter Wong**

School of Property, Construction and Project Management, RMIT University, 124 La Trobe St, Melbourne, VIC 3000, Australia; tayyab.maqsood@rmit.edu.au (T.M.); Malik.khalfan@rmit.edu.au (M.K.); rebecca.yang@rmit.edu.au (R.J.Y.); peterspwong@rmit.edu.au (P.W.)

\* Correspondence: salman.shooshtarian@rmit.edu.au

**Abstract:** With increased construction activities in capital cities of Australia, the sustainable management of construction and demolition (C&D) has become an important item in the federal and state government agendas. According to the universally accepted concept of waste hierarchy waste disposal is the worst preferred waste management option due to environmental issues. Currently, in most Australian jurisdictions, a landfill levy is applied to discourage waste disposal and to further encourage waste recovery. However, there is an ongoing debate as to whether the levy regime could achieve the desired outcome. Therefore, this study, funded by the Australian Sustainable Built Environment National Research Centre, explored the effectiveness of the current landfill levy across Australian jurisdictions. The paper presents the findings of this study that were obtained from a questionnaire survey aiming to capture the main C&D waste management stakeholders on landfill taxing imposition in Australia. The study collected 132 responses from professionals in the construction industry and other industries dealing with C&D waste management and resource recovery. The results demonstrated that those who believed in market incentive approaches outweigh people that were in favour of pecuniary impost approach. Among those who favoured pecuniary imposts, almost 90% of participants agreed with the effectiveness of landfill levies in any waste management system. Other results provided a useful insight into the actual implications of the current levy scheme. It is expected that the findings in this study contribute to developing sound policies that provide a level field for all key stakeholders and to ensure that resource recovery is further encouraged.

**Keywords:** construction activities; legislation; waste management; stakeholders; resource recovery

## 1. Introduction

Construction activities in Australian cities have substantially grown over the recent decades leading to the generation of a large amount of waste [1]. Construction and demolition (C&D) waste stream, therefore, accounts for 43% of the total waste generated, reaching 20.4 Mt annually. The average annual growth of C&D waste generation is currently at 2%, and about 6.7 Mt of this waste stream is landfilled. According to the universally accepted concept of waste hierarchy, waste disposal is the worst preferred waste management option due to environmental issues [2]. Despite having no physical shortage of landfill sites, there is a high level of community environmental awareness; meaning that it is difficult to open new landfill sites and incineration is also not tolerated [3]. To add to the complexity, recent anti-waste movements by foreign countries such as China, Thailand, Philippines and Malaysia that ban the import of recyclable from developed nations [4,5] has left Australia with alarming rates

of waste stockpiling [6,7]. This is mainly because the waste producers can no longer avoid landfill levies or recovery operation fees by shipping waste overseas. Although this policy only focuses on certain types of metals, textiles, plastic and cardboards used in packaging and not all C&D waste, the announced level of acceptable contamination is a real hurdle to the export of C&D waste from Australia. Some Australian organisations have claimed that the ban diminishes the ability of material recovery facility (MRF) operators to market sorted recyclables, and consequently stockpiling and more landfilling is likely to occur [8].

As such, state governments attempt to regulate C&D waste management by enforcing relevant legislation. The average annual growth rate of C&D waste recovery is estimated to be 3% from 2007 to 2017. However, the effectiveness of these regulations is still debatable [9–13]; the key stakeholders in the construction and waste management industries, along with proven evidence, have questioned the actual impact [8]. For instance, while Australia National Waste Report 2018 indicated that the annual C&D waste recycling rate is 13.6 Mt or 67%, holding the largest rate compared to other waste streams [14], there is proof of substantial stockpiling, illegal dumping and landfilling of C&D waste in Australia [14].

It seems that multiple factors contribute to the emergence of this contradictory situation. These factors range from government regulation through to available financial incentives or barriers. Among these factors, the landfill levy is a pecuniary impost that has been imposed on waste landfilling for several years [15]. The levy aims to set a price on waste disposal that is higher than the cost of recycling, such that recycling becomes a more attractive endpoint. Furthermore, while appropriately designed levies can enable waste recovery activities such as energy recovery from waste materials [10,16] which would not have been commercially competitive against landfill without levies, these activities would not be [17].

In Australia, the landfill levy rate is determined by state and territory governments [18], and depending on the factors involved in the formulation, it differs from one state to another [1]. The first-ever implemented Australian levy scheme was enforced in New South Wales (NSW) metropolitan area in 1971 [19], which was rated at $0.51 per tonne. A scheme that was subsequently followed by some other Australian states and territories. Except for Northern Territory (NT) and Tasmania (Tas), all of Australia's other jurisdictions (Australian Capital Territory (ACT), NSW, Victoria (Vic), Queensland (Qld), South Australia (SA), Western Australia (WA)) have introduced landfill levies. The sources of disparity in landfill levy imposition are the location of waste disposal being metropolitan or regional, being obligatory versus voluntary, providing a tax exemption for certain materials and having levy zones. At the time of writing, ACT has set the largest levy rates (metropolitan: $146.2, regional: $199.2) among Australian jurisdictions, followed by NSW (metropolitan: $141.2, regional: $81.3) and SA (metropolitan: $140, regional: $70). The main factors giving rise to non-uniform levy rates include state revenue requirements and priorities, cost disparity between jurisdictions and regional versus metropolitan areas, available facilities and regulatory framework.

Technically, a part of revenue earned from waste disposal is used to improve enforcement and compliance, development of sound policies, and to fund actions and strategies that contribute to waste minimisation. In Australia, there is no nationally prescribed method for distribution of levies for such purposes, and each state government does so according to its priorities and the objectives. Among jurisdictions that charge levies, only Vic, NSW and WA have a firm plan that determines how the levies charged at landfill sites should be distributed. In other jurisdictions, the funding has been granted on a case-by-case basis and according to waste jurisdictional strategy objectives. In 2018–2019 landfill levy generated $1.13 billion revenue across Australia, of which $282 million (25%) nationally was reinvested into activities pertaining to waste and recycling, state Environmental Protection Agencies (EPAs) or climate change [17].

The cost disparity that exists between Australian jurisdictions, and between regional and metropolitan areas were another source of disagreement. Driven by various factors such as facility availability, regulations, economic benefits and common industry practices, it has been reported

that C&D waste is being transported long distances from originally generated location to a place of disposal [11,18,20].

While in some circumstances a landfill levy is the best economic driver, it can act as a disincentive in other circumstances. Previous literature reported conflicting results about the effectiveness of landfill levy, both in domestic and international contexts. For instance, in Hong Kong, a three-year levy scheme (2006–2008) demonstrated that C&D-specific waste levy taxes could influence construction's behaviours regarding C&D waste, resulting in a significant reduction in solid waste disposal [21]. In 2011, a C&D supply chain guide [22] prepared for the Commonwealth Government of Australia reported that many stakeholders had indicated that landfill costs (landfill operation and levies) are a significant driver for the use of salvaged and recycled C&D waste. A recent report form WA's government [23] revealed that the increases in the waste levy have resulted in a decline in waste disposal from the Perth metropolitan, particularly in C&D waste stream; however, due to the lack of reliable data, it is not clear if this trend has led to increased waste recovery activities. On the contrary, some evidence presented in an Australian parliament document suggested a failure in achieving the intended goals (e.g., reduction in waste disposal or an increase in waste recovery activities) of imposing a landfill levy [8,13]. Another Australian based study [12] found that an increase in the levy rate not only did not solve the problem but could also lead to illegal dumping. Indeed, the evidence demonstrated the limits to what can be achieved through the imposition of a landfill levy.

There is limited research around the acceptability and perceived impact of C&D waste landfill levy on the waste and resource recovery industry among key stakeholders in the Australian context. Therefore, this study aimed to capture the key stakeholders' perceptions of the landfill levy imposition status quo in Australia, including its effectiveness in reducing C&D waste landfilling. Two specific objectives of this study were:

1. To explore C&D waste key stakeholders' perceptions of the landfill levy.
2. To identify the best approach to improve the effectiveness of current landfill levy regimes.

The study is part of a larger national study (Project 1.65. A National Economic Approach to Improved Management of Construction and Demolition Waste) that investigated the harmonisation of C&D waste management systems in Australia. The study was funded by the Australian Sustainable Built Environment National Research Centre (2018–2020).

## 2. Methodology

### 2.1. Sample and Data Collection

A cross sectional survey of a random sample of stakeholders of C&D waste management operating in different jurisdictions of Australia was conducted from June to September 2019. They were recruited through the project working industry and government partners including Waste Management and Resource Recovery Association of Australia and Australia Sustainable Built Environment National Research Centre. An email, including the online link to the survey and the information sheet, was sent to 250 individuals inviting them to participate. A similar email was sent as a reminder about one month later, to the same members. Participation in this study was voluntary and a completed survey implied informed consent. The investigators maintained the privacy and confidentiality of all information collected in the study as per the Australian National Statement on Ethical Conduct in Human Research and RMIT University Human Ethics Committee, College Human Ethics Advisory Network (CHEAN) who approved the survey content and administration (CHEAN A&B 21847-11/18).

### 2.2. Survey Design

Development of the survey involved review of the literature, regulations, policy statements, other international guidelines and expert opinions regarding the main issues of C&D waste management in Australia. Based on this information, an online survey was designed using Qualtrics [24]

an online survey toolkit typically used in social sciences studies [25]. The questions in the survey were presented in 11 domains which included demographic details, general overview of C&D waste management issues, levy imposition, C&D waste regulation, the impact of (and response to) China's new waste policy, waste-related advisory/mandatory schemes, C&D waste management practice in construction firms, C&D waste material recycling in recycling facilities, C&D waste and construction material manufacturers, C&D waste and delivery and transport, and establishment of C&D waste marketplace. The survey design consisted of multiple-choice questions (e.g., 'true'/'false'/'not sure'), 7 point Likert scales (1 = 'strongly agree' to 7 = 'strongly disagree'), and single and multiple text entries in ranked order (1 to 5). The completed survey was reviewed by experts to establish content validity and then tested for online accessibility and comprehension by a group of construction professionals and researchers. The content, clarity and length of the survey was then modified accordingly. Furthermore, a series of questions were presented to the participants seeking their opinions about the validity of the survey.

The final version of the survey contained 46 questions which were reduced according to the responses recorded using the Qualtrics' "logical skip" function that was set up to present certain questions that were relevant to a specific group of respondents. The average response time was recorded to be between 25 and 30 min.

This study only focused on two domains (i.e., participants' demographic details and landfill levy). In total, these two domains consisted of nine questions that were designed to elicit the most relevant information about the quality of landfill levy imposition in Australia. The questions on participants' demographic details sought the participants' field of activity, experience, and the primary location of their activity. Questions in the landfill levy domain focused on participants' opinion on the best approach to reduce the volume of waste going to landfill, and the effectiveness of landfill levy regimes and their characteristics.

### 2.3. Data Analysis and Presentation

In total, 132 responses were received and recorded in the Qualtrics database. After screening the responses, the data from the survey was analysed using Excel Spreadsheets 2016. To explore the participants demographic details and their opinion on the landfill levy descriptive statistical analyses were used. Frequency distribution was the main statistical measure to compare different categories of responses received from participants. In the case of text entry questions, the responses were combined and categorised for analytical purposes. Excel Spreadsheets 2016 was also used to present the results in the form of graphs and tables.

## 3. Results

### 3.1. Participants Profile

To better understand and analyse the responses received from the participants three questions were formulated to seek participants' demographic data, including the industry and geographical zone that they perform their main activities and the length of experience. More than 50% of the participants belonged to three sectors: waste recovery (20%), construction (16%), and landfilling (15%). In terms of the length of experience working in C&D waste space, as shown in Table 1, about 44% of participants had less than six years, and only less than 30% of them worked in an industry related to C&D waste management more than 15 years at the time of running the survey. More than 60% of participants were based in two large states of Vic and NSW (Table 1).

**Table 1.** Study participants' profile.

| Question | Distribution | (%) |
|---|---|---|
| **Field of activity** | Construction | 16 |
| | Demolition | 8 |
| | Landfill | 15 |
| | Legislation | 6 |
| | Industry association | 6 |
| | Waste recovery | 20 |
| | Waste delivery and transport | 10 |
| | Consultancy | 7 |
| | Manufacturing | 4 |
| | R&D | 3 |
| | Regulations & enforcement | 5 |
| **Experience** | <6 years | 43.1 |
| | 6–10 years | 13.7 |
| | 11–15 years | 16.7 |
| | >15 years | 26.5 |
| **State/territory** | ACT | 1.8 |
| | NSW | 24.3 |
| | NT | 6.3 |
| | Qld | 16.2 |
| | Tas | 3.6 |
| | Vic | 30.6 |
| | WA | 17.1 |

Australian states/territories: Australian Capital Territory (ACT), New South Wales (NSW), Northern Territory (NT), Queensland (Qld), Tasmania (Tas), Victoria (Vic), and Western Australia (WA).

### 3.2. Approaches to Reduce C&D Waste Landfilling

In waste-related literature and policies, the main three viewpoints underpinning landfill waste reduction activities are encouragement, enforcement and education. In the context of Australian C&D waste management, each of these has its own set of advantages and disadvantages, and hence received a different level of support and criticism. The survey targeted the key C&D waste stakeholders' perception about the best approach. Two questions were posed to determine the extent to which participants support a. more market incentives (encouragement) and b. more pecuniary imposts (enforcement). The questions were:

a.　Express your degree of agreement on the following statements: More market incentives (e.g., remove regulatory barriers; foster minimum recycled content in products; invest in technologies to recycle and facilitate trade such as trading platforms) can better increase C&D waste reduction, re-use and recycling in Australia.

b.　Express your degree of agreement on the following statements: More pecuniary imposts (e.g., landfill levy increases; taxes on producers; more regulations, monitoring and enforcement) can better increase C&D waste reduction, re-use and recycling in Australia.

For the first category, the responses showed that the respondents significantly favoured market incentives (Figure 1), with more than 92% indicating agreement or somewhat agreement. The more pecuniary approach received significant but comparatively less support; the results showed that about 70% of participants indicated agreement with implementing more pecuniary imposts given enforcement activities status quo (Figure 2).

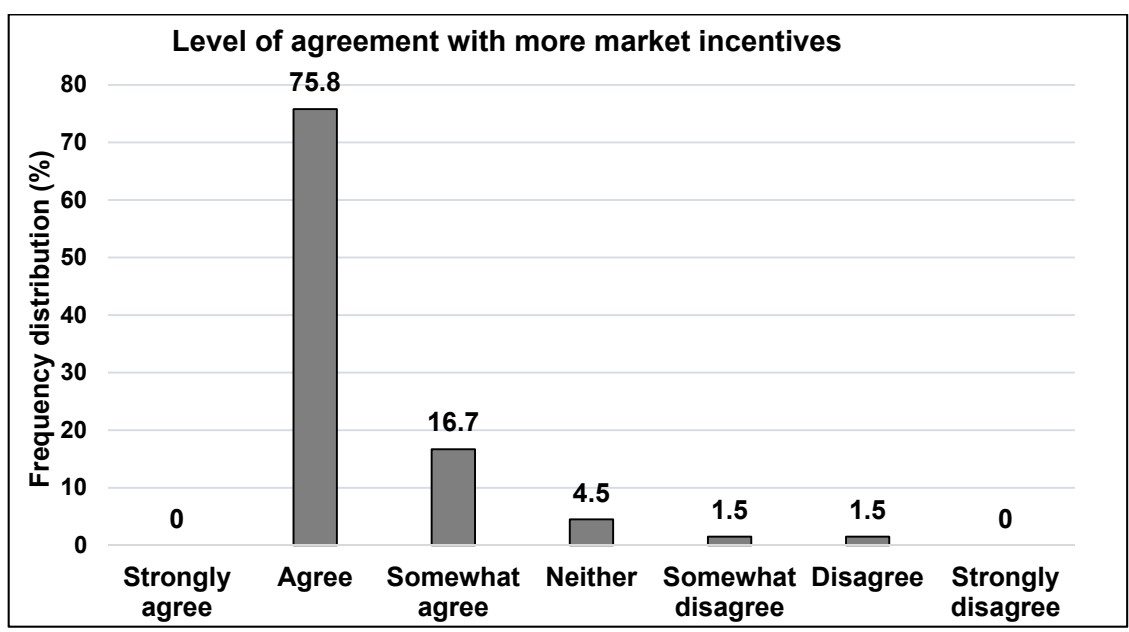

**Figure 1.** Frequency distribution of agreement level with the increased implementation of market incentives in construction and demolition (C&D) waste stream.

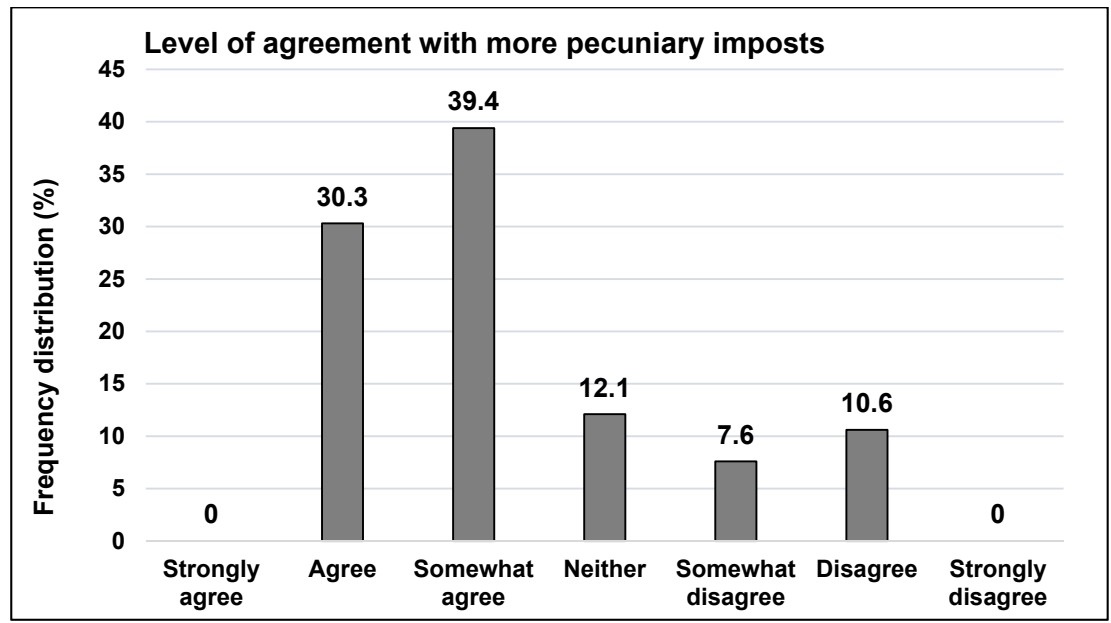

**Figure 2.** Frequency distribution of agreement level with increased implementation of pecuniary imposts in C&D waste stream.

### 3.3. Effectiveness of Landfill Levies in C&D Waste Management Frameworks

There are some doubts about the position of landfill levies in waste management systems and how they contribute to promoting C&D waste minimisation and resource recovery. Therefore, the respondents were also asked to express their agreement/disagreement level with a statement about the effectiveness of landfill levy. The results showed that almost 90% of participants endorsed the effectiveness of the landfill levy in general (Figure 3). Therefore, it can be concluded that in the participants' opinion, the landfill levy is an integral part of any C&D waste management system.

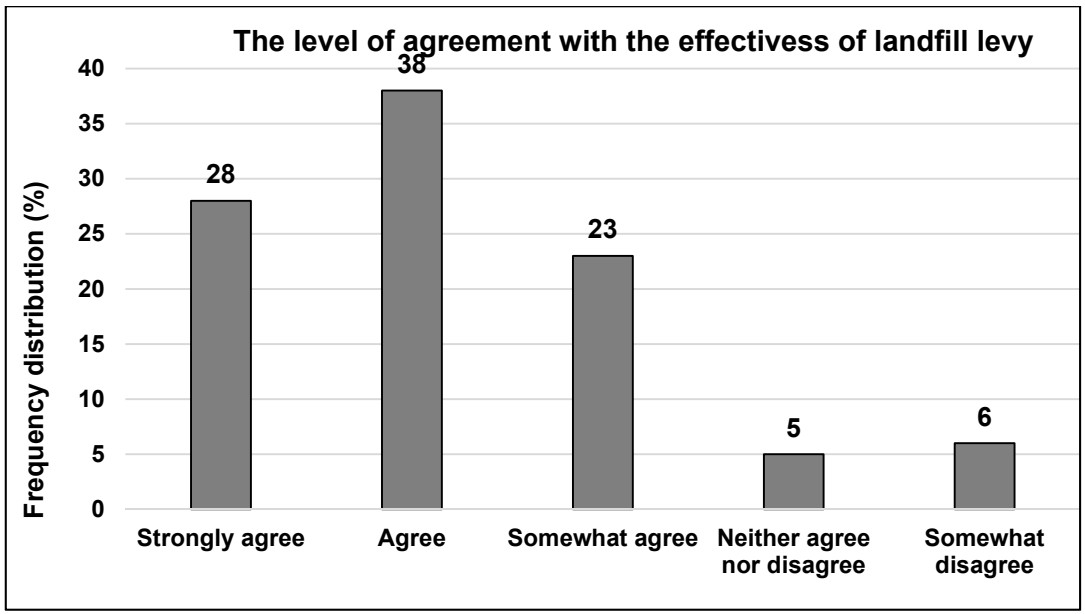

**Figure 3.** Frequency distribution of the agreement with the effectiveness of the landfill levy.

*3.4. Characteristics of the Landfill Levy*

To further understand the adequacy of the application of the current landfill levy regime enforced in their jurisdiction, the participants were asked to provide a descriptive response to a series of questions seeking their opinion on landfill levy characteristics. These questions were only provided to those who previously indicated that a landfill levy is an effective way to reduce the volume of waste going to landfill. Their responses were categorised based on the highest frequencies and are presented in multiple graphs below (Figure 4).

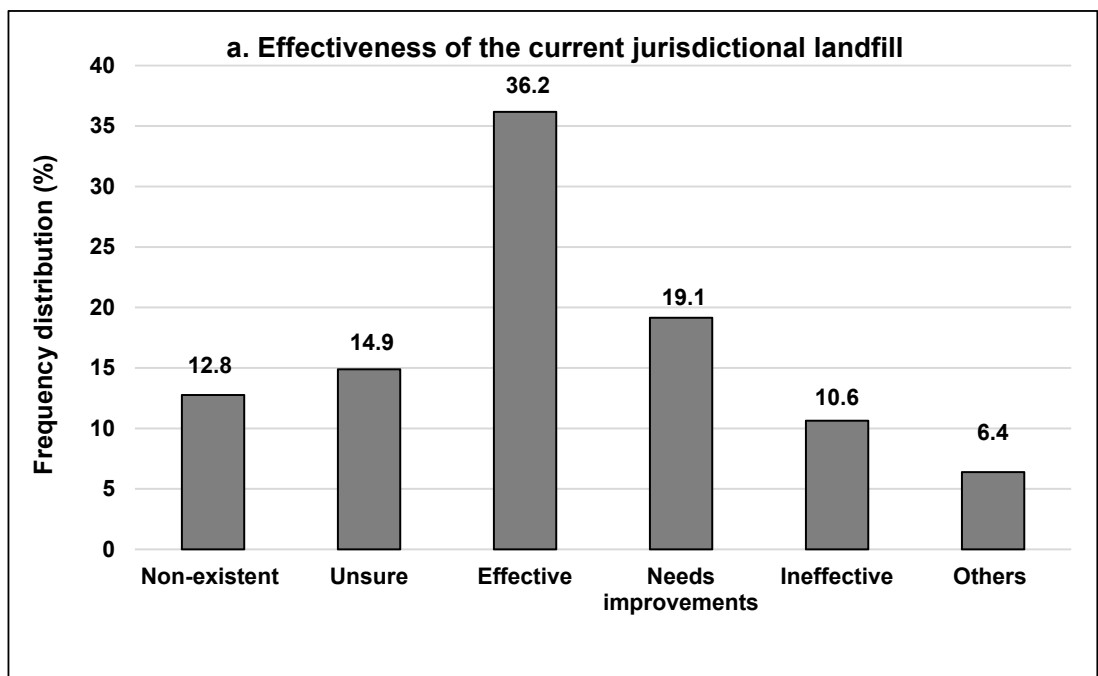

**Figure 4.** *Cont.*

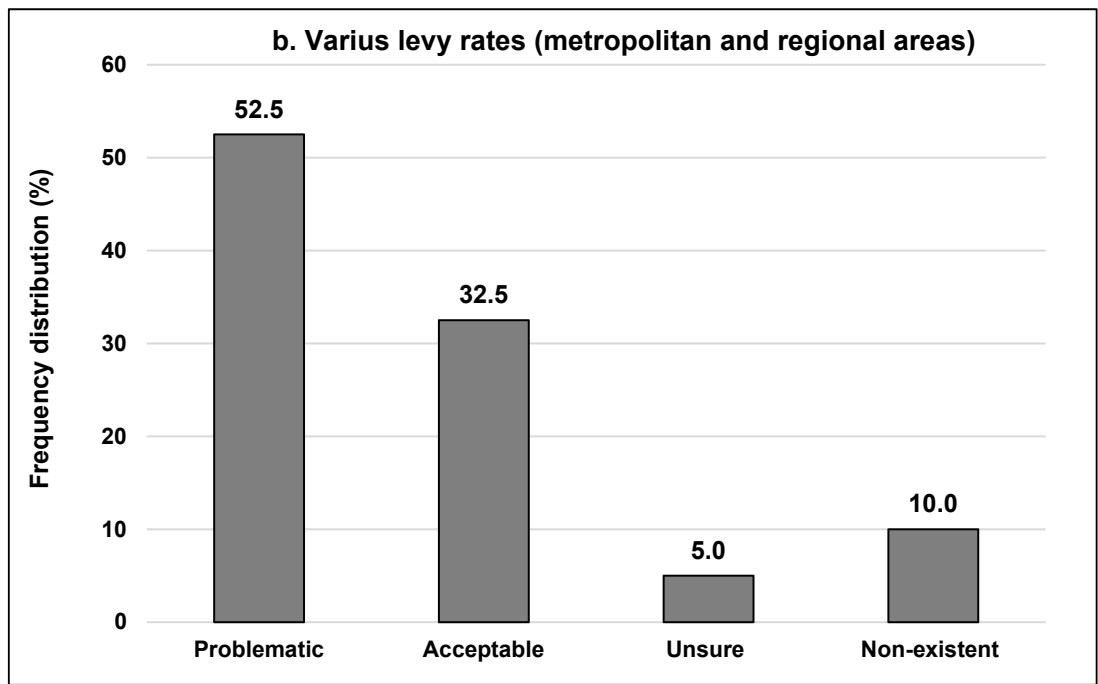

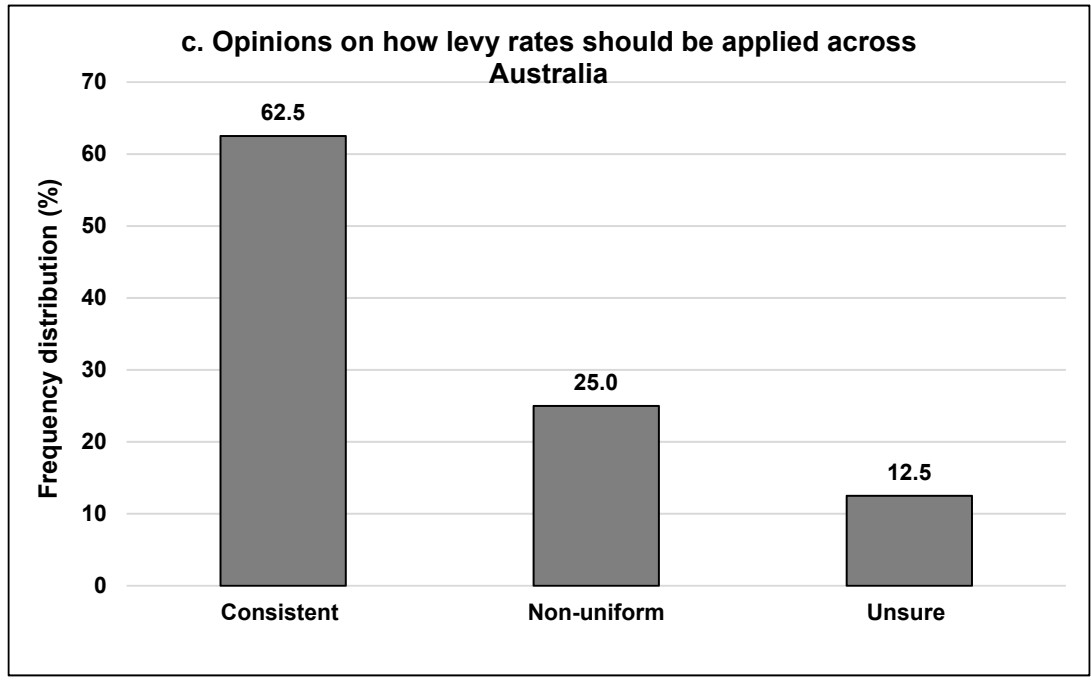

**Figure 4.** *Cont.*

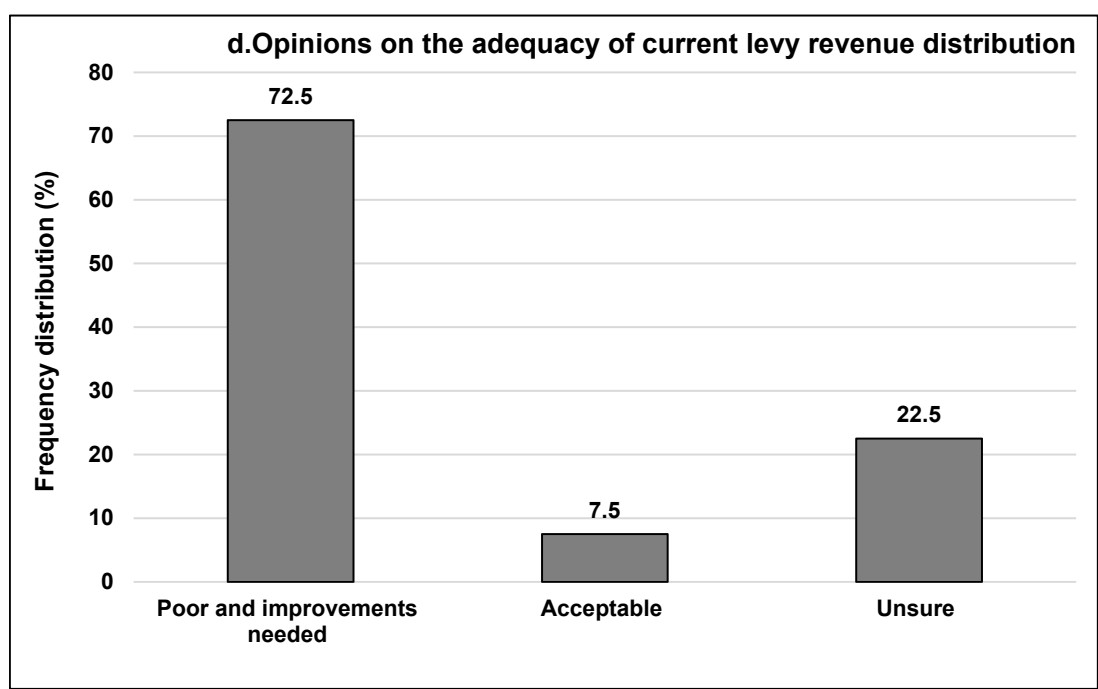

**Figure 4.** Participants responses frequency distribution (%) to questions related to (**a**) the effectiveness of landfill levy, (**b**) levy rate differential rate between metro and regional areas, (**c**) consistency of levy rate across Australia, and (**d**) distribution of levy revenue.

The questions required participants to comment on the effectiveness of current landfill levy specifically imposed in their jurisdiction (Figure 4a), the levy rate difference between metropolitan and regional rates (Figure 4b), the consistency of landfill levy imposition across Australian jurisdictions (Figure 4c) and adequacy of levy revenue distribution (Figure 4d).

On the effectiveness of jurisdictional landfill levy, 36.2% of participants endorsed that the current levy has a moderate to full impact on effective C&D waste management, while 29.7% mentioned the current levy is not effective and needs improvements. The rest were either unsure about its effectiveness (13.6%) or reported that they did not have levy schemes in their jurisdiction (12.8%) at the time of survey, meaning they were based in NT or Tas (Figure 4a).

In some Australian states, there is a discrepancy in the landfill levy rate between regional and metropolitan areas. This cost disparity has been reported to be both positive and negative [8] and can be the source of a few issues, including waste transport or illegal waste dumping. In response to a question related to this cost disparity, participants revealed varying opinions (Figure 4a). About 52.5% of participants believed that different levy rates are problematic, and consistency is needed to avoid unintended consequences and unnecessarily complications. However, about 32.5% of participants indicated that differences in levy rates between metro and regional areas are practical and effective; they even stated that further discount and waivers are needed for regional areas (Figure 4b). Hence, further investigation is needed to fully understand how discrepant levy rates can act as a powerful tool for waste management or otherwise.

Cost disparity in levy rate also exists between Australian states. The participants were asked to indicate their opinion about this cost disparity. The results showed an overwhelming majority of responses (62.5%) supported having a national consistent levy rate (Figure 4c). About a quarter of responses supported non-uniform levy rates.

Lastly, there is a debate on how to spend the revenue generated from landfill levy imposition. Therefore, this survey targeted respondents' opinion about the current system of levy revenue distribution. The results, as shown in Figure 4d, suggested that majority of participants (72.5%) believed that there is a poor distribution of the levy revenue and improvements needed to maximise

waste recovery in C&D waste sector. One widely referenced recommendation was to return the revenue to the waste industry. The other interesting finding was that almost one-fifth of participants were unsure about the current distribution system.

## 4. Discussion

### 4.1. Improvements

As shown in the results, the key stakeholders were agreeable to more pecuniary imposts (Figure 2) and among those who agreed to it some were critical about the current levy regime in Australia despite knowing that it is an effective tool to manage C&D waste in general (Figure 4a). In response to questions related to levy characteristics, several recommendations were made by the participants to improve its application across Australia. Among other recommendations, national harmonisation of gate fees was repeatedly indicated. In previous studies, it was argued that a national gate fee could minimise interstate waste transfer [11,18] and reduce complications for companies operating in various states [8,13]. Particularly, in this study, one participant mentioned that "[levy rate disparity] encourages waste movement vehicle accidents increased carbon emissions—just moving the issue 200 km away".

However, it should be noted that such an arrangement might not necessarily produce the best results. Simple harmonisation may overlook the existing contextual conditions in each jurisdiction. It may also interfere with the specific waste management systems planned and implemented in different states and territories. Below are some examples of participants' opinion comments in favour of the levy rate difference between regional and metropolitan areas:

- "[Levy rate] needs to be higher in regional areas to promote recycling. Otherwise there is not the market for C&D material in these locations".
- "Levy rates should be less in regional (Qld) where there is no option to recycle".
- "Agreed due to the remoteness conditions of the NT, which could help preventing illegal dumping".
- "A differential levy works in mainland states because of the vast geographic size of mainland states".
- "Regions are too diverse to be able to manage the same way as metro areas due to lack of resource availability".

Hence, it is better to set up the levy fees in a way that the negative impact on the effective management of C&D waste across Australia is minimised. For instance, if there is to be a rate disparity, some precautionary measures should be considered to make sure that unnecessary long-distance waste transfer is avoided. Another point of debate was about the right rate for the landfill levy that produces the best possible results. While many participants appreciated the positive impact of the increased levy rate on waste recovery and supported the regular increase in the rate, others stated that raising the rate may induce unintended consequences. These could include illegal dumping and stockpiling, waste transfer, increased economic pressure on recyclers due to high levy, poorer quality of recyclable material entering the market and driving up the cost of treatment, changes to the market (market distortion), increased building costs and reduce housing affordability. Below are some of the arguments provided by respondents:

- "It is ineffective in WA due to [Department of Water and Environmental Regulations] DWER being unable to control illegal operations".
- "It increases building costs without reducing waste".
- "As per findings by the Waste Authority in 2014 before the levy was increased, the cost of transport is a key factor in providing incentive to recycle rather than landfill. But if you are generating waste next to a landfill, it is cheaper to landfill than recycle in WA".

As a result, it is suggested that any increase in levy rate should be complemented with effective compliance and enforcement regime that relies on technologies such as a global positioning system

(GPS) tracker. Furthermore, it is noteworthy to consider transport costs and potential cost implications for construction activities when determining the levy rate.

Lastly, the comments on the levy revenue suggested the need for allocating the revenue to the activities that directly improve the waste and resource recovery industry. For instance, levy revenue could be used to invest in the development of a market for recycled materials through low-interest (subsidised business) loans or financial incentives and research and development (R&D). In Australia, currently, there is no nationally prescribed method for distribution of levies for such purposes, and each state government does so according to its priorities and the objectives [17]. For instance, the NSW state government has invested $900 m over a nine-year period (2010–2020) from landfill levy revenue in the waste management and resource recovery sector. This investment is made in light of a "waste less, recycle more" funding initiative [26] that outlines the state reinvest levy revenue plan and objectives. In Vic, in 2010, the Environmental Protection Agency (EPA) enacted regulations [27] to determine the share of levy revenue to be received by each state agency. In this state, during the 2015 financial year, the levy revenue was distributed to the following agencies: the EPA received 18%, Sustainability Victoria received 12%, and the Waste and Resource Recovery Groups (Councils) received 5%. Similar to Vic, WA has a piece of primary legislation that specifies the arrangement of levies imposition and distribution. This legislation outlines how the money that is collected is used by the Environment Minister to (a) fund certain programs relating to the management, reduction, reuse, recycling, monitoring or measurement of waste and (b) fund a person or body to conduct the activities mentioned in (a).

As noted before, however, other state governments have not revealed a firm plan that determines how the levies charged at landfill sites should be distributed. In this situation, funding has been granted on a case-by-case basis and according to waste jurisdictional strategy objectives. In SA, the waste strategy (2015–2022) declared that more than $80 million sourced from levy revenue had been invested into the industry during the past decade; the fund allegedly was used by Zero Waste to build capacity and improve industry competitiveness (South Australia's Waste Strategy, 2015–2020 (p. 10)). This strategy also indicated that 50% of the levy should be transferred to the Waste to Resources Fund under the Zero Waste Act 2004, to be spent in accordance with the Zero Waste SA Business Plan. In Qld, In 2019–2020, it is estimated that of $1.54 billion in funds raised, around $569 million or 37% will be reinvested into waste and recycling activities [17].

### 4.2. Government Role

In Australia, state and territory governments have a pivotal role in the management of C&D waste, including design, implementation and monitoring of landfill levy. Government agencies who are responsible for the imposition of the landfill levy should carefully consider several factors in the design of the landfill levy regime to avoid potential issues. One study in Qld showed how unprecedented state government decision in introducing and revoking of landfill levy significantly impacted the recycling rate of C&D waste in this state [9,28].

Furthermore, state and territory governments are responsible for providing financial support for waste management and resource recovery activities in agreement with objectives stipulated in state waste strategy documents [29]. Government funding to improve waste and resource facilities together with effective law enforcement provides an impetus for further waste recovery activities. An increase in the number of local infrastructures discourages waste producers and collectors (waste responsible) from sending waste across the Australian states such that it would be easier to implement the proximity principle. Technically, a lot of waste minimisation practices and strategies, such as extended producer responsibility, depend on the availability of technologically advanced local infrastructures.

As noted before, the typical source of these supports is the revenue derived from landfill levy imposition [18], which is not always eventuated as planned. In 2016, Ritchie (cited in [28]) indicated that Australian federal, state and territory governments develop waste management strategies, but then do not bother to create accompanying favourable economic and policy conditions to allow businesses

and local governments to achieve these targets. Serpo and Read [17] suggested the following reforms to maximise the benefit of revenue investment in the waste-related industries:

- Maintain a separate waste levy trust account from which all levies collected are managed;
- The Trust Account should have clear rules on how the funds are to be allocated and reported on, including objectives that link to the State's waste avoidance, resource recovery plans;
- Levies raised are only invested in activities consistent with the Trust Account's rules and objectives;
- Guaranteeing a minimum percentage of levies to be spent annually on activities to implement the jurisdiction's waste avoidance and resource recovery strategies, resource recovery and remanufacturing industry development plans, market development initiatives and infrastructure plans.

The federal government in particular, as agreed in the Environment and Communications References Committee [8], should take leadership in the development of national waste policies that direct waste recovery-related activities such as levy design and imposition with the aim of consistent management of C&D waste across states and territories.

### 4.3. Market Development

Despite the preventative impact of landfill levy on waste disposal, market incentives were found to be a complementary strategy in the management C&D waste space. Development of the domestic market for recycled materials is an effective approach that is repeatedly recommended in policies, guidelines and previous studies [13,29–31] given the fact that overseas waste destinations are no longer available to Australian C&D waste-related industries [6]. The successful development of a C&D waste material market hinges on several factors. Shooshtarian, Maqsood, Wong, Khalfan and Yang [13] identified seven main factors in the development of C&D waste marketplace that included regulatory support, design and implementation of extended producer responsibility (EPR) scheme, the establishment of the effective and integrated supply chain, sustainable procurement scheme, investments in technology and infrastructure, research and development and properly designed landfill levy imposition. On the same note, Caldera, Ryley and Zatyko [31] stated that market-based policy instruments could be developed through taxes, subsidies and other incentives, to encourage waste diversion from landfills, recycling and creating a second life for waste material. To market the recycled material as a substitute for natural raw materials, it is essential to increase awareness and carry out promotional activities. Then a continuous supply of clean waste streams is necessary to produce high-quality recycled content that satisfies the given technical specifications and is economically competitive.

### 5. Conclusions

The issue of C&D waste management has long been on the Australian federal and states governments' agenda. Therefore, policymakers have started enquiring into the effectiveness of current policies, such as landfill levy schemes, in alleviating C&D waste issues. In doing so, they need to ensure that any changes in relevant policies will achieve maximum consensus among stakeholders and will not leave unreasonably negative impacts on those who are involved in C&D waste management and resource recovery activities. Therefore, the present study particularly aimed to capture the perceptions of the main C&D waste management stakeholders on the current operation of landfill levy across Australia. The study successfully provided critical information on the issues around landfill imposition in the C&D waste stream. In total, 132 participants from across Australia took part in the online survey and contributed to our understanding of the effectiveness of current landfill levy schemes imposed individually in different Australian jurisdictions.

On the one hand, results showed that about 70% of participants had a level of agreement with implementing more pecuniary imposts and indicated that the landfill levy is generally a useful tool to promote C&D waste minimisation and resource recovery. Therefore, it can be confirmed that the levy

is an integral part of the C&D waste management system. On the other hand, a higher proportion of votes (90%) was recorded for more market incentives. These two pieces of information reflect the fact that participants perceived pecuniary imposts and market incentives as complementary approaches in forming an effective waste management system rather than opposing strategies. The results suggest that Australian waste management frameworks need to be geared towards more balanced regulation of waste streams, where stick and carrot measures are proportionately considered.

Furthermore, four characteristics of the landfill levy in Australia, including the effectiveness of current Australian landfill levy regimes, the rate difference between metro and regional areas in one state and between jurisdictions, and allocation of revenue generated from landfill levy to the waste-related industries, were investigated. The results showed that stakeholders believed that the current landfill levy schemes implemented in Australia are not as efficient as they should and need improvements. This study explored opportunities for improvements and provided relevant recommendations accordingly. The recommendations involved making an informed decision on harmonisation of landfill levies across Australia, complementing levy imposition with effective compliance and enforcement regimes that rely on technologies such as a global positioning system (GPS) tracker, consideration of transport costs and potential cost implications for construction activities when determining the levy rate, reinvestment of landfill levy revenue in resource recovery activities through, for example, providing low-interest loans or financial incentives, supporting research and development (R&D), and increasing the number of local infrastructures.

Furthermore, the results indicated that market development is a sustainable solution to solve some of C&D waste issues. Relevant factors influencing the operation of such a market were identified as government regulatory support, design and implementation of EPR scheme, the effective supply chain, sustainable procurement scheme, investments in technology and infrastructure, R&D and imposition of properly designed landfill levy scheme. It is expected that the findings in this study contribute to developing sound policies that provide a level field for all key stakeholders such as construction and demolition firms, waste collection businesses, recycling facilities, landfill owners, government sector and the general public. This level field should eventually and further resource recovery activities.

**Author Contributions:** Conceptualization, S.S., T.M., R.J.Y., M.K. and P.W.; methodology, S.S., T.M., P.W.; formal analysis, S.S.; writing—original draft preparation, S.S.; writing—review and editing, S.S. and T.M.; project administration, S.S., T.M.; funding acquisition, T.M., M.K., P.S.P. and R.J.Y. All authors have read and agreed to the published version of the manuscript.

**Funding:** This research was funded by the Australia's Sustainable Built Environment National Research Centre, grant number [p. 165].

**Acknowledgments:** The authors would like to acknowledge the support of the Australia's Sustainable Built Environment National Research Center, Project 1.65. However, the views expressed in this article are those of the authors and do not necessarily represent the views of SBEnrc.

**Conflicts of Interest:** The authors declare no conflict of interest.

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
