# Peer review of "Landfill Levy Imposition on Construction and Demolition Waste: Australian Stakeholders’ Perceptions"

_sustainability, doi:10.3390/su12114496_

Round 1
Reviewer 1 Report
- The sentence corresponding to the line 55 seems to be not complete
- It should be interesting if the authors would present the structure of the entire questionnaire and how all the questions have been selected, explaining why the six questions selected for the study are more relevant than the others according to the research objectives.
- It is not clear the difference between the two graphs in figure 1, different data are showed with the same didascaly
- There are no didascalies to the graphs at pages 6 and 7 that enable to interpret the results showed.
- The data expressed in figures at pages 10-14 should be better commented
Author Response
|
1 |
The sentence corresponding to the line 55 seems to be not complete |
We are not sure if we correctly located the sentence corresponding to the line 55 due to the change in the format of the manuscript, however, based on the newly formatted manuscript we have revised the sentence no.55:
“The first-ever implemented Australian levy scheme was enforced in New South Wales (NSW) metropolitan area in 1971 [11], which was rated at $0.51 per tonne”. |
|
2 |
It should be interesting if the authors would present the structure of the entire questionnaire and how all the questions have been selected, explaining why the six questions selected for the study are more relevant than the others according to the research objectives. |
We have now added the structure of questionnaire to the methodology as per the comment:
The online questionnaire consisted of 11 question packages with each package being intended for investigation of a particular C&D waste management dimension in Australia. These include participant demographic details, general overview of C&D waste management issues, levy imposition, C&D waste regulation, the impact of and response to China’s new waste policy, waste-related advisory/mandatory schemes, C&D waste management practice in construction firms, C&D waste material recycling in recycling facilities, C&D waste and construction material manufacturers, C&D waste and delivery and transport, and C&D waste marketplace. For this study, only responses from two packages, demographic details and landfill levy, were selected and analysed. In total, nine questions that provided the most relevant information about landfill levy imposition and the current research study were used in this study. These questions covered the participants’ demographic details (i.e. field of activity, experience and location of activity), , participants’ opinion on the best approach to reduce the volume of waste going to landfill, the effectiveness of landfill levy regimes, and characteristics of landfill levy |
|
3 |
It is not clear the difference between the two graphs in figure 1, different data are showed with the same didascaly |
We have separated the two graphs and they are now captioned as Figure 1 and Figure 2. The description of these two graphs are provided below:
The responses showed that the respondents are more in favour of market incentives, with more than 92% level of agreement and somewhat agreement (Figure 1 ). On the contrary, as shown in Figure 2, about 70% of participants had a level of disagreement with implementing more pecuniary imposts. |
|
4 |
There are no didascalies to the graphs at pages 6 and 7 that enable to interpret the results showed. |
We revised the description of results presented in graphs 1, 2 as follows:
In waste-related literature and policies, the main three viewpoints underpinning landfill waste reduction activities are encouragement, enforcement and education. In the context of Australian C&D waste management, each of these has its set of advantages and disadvantages, and hence received a different level of support and criticism. To understand how the key C&D waste stakeholders position themselves concerning increasing C&D waste landfilling reduction, a question was posed that sought their opinion about the best approach. The question required participants to indicate whether they support a. more market incentives (encouragement) and b. more pecuniary imposts (enforcement) . For the first category, the responses showed that the respondents significantly favoured market incentives (Figure 1), with more than 92% level of agreement and somewhat agreement (Figure 1 ). On the contrary, , more pecuniary approach was not supported; the results showed that about 70% of participants had a level of disagreement with implementing more pecuniary imposts given enforcement activities status quo (Figure 2).
For graph 3, below is the description:
The respondents were also asked to express their agreement/disagreement level with a statement about the effectiveness of current landfill levy to promoting C&D waste minimisation and recovery. The results showed that almost 90% of participants endorsed the effectiveness of landfill levy. Therefore, it can be concluded that in the participants’ opinion, the landfill levy is an integral part of Australian C&D waste management system in the general sense.
|
|
5 |
The data expressed in figures at pages 10-14 should be better commented |
We improved commentary on figure 4 as follows:”
On the effectiveness of jurisdictional landfill levy, 36.2% of participants endorsed that the current levy has a moderate to full impact on effective C&D waste management, 29.7% mentioned the current levy is not effective and needs improvements. The rest indicated whether they are simply unsure about its effectiveness (13.6%)or reported that they do not have levy schemes in their jurisdiction (12.8%), meaning that participants were based in NT or Tas (Figure 4.a).
In some Australian states, there is a discrepancy in the landfill levy rate between regional and metropolitan areas. This cost disparity has been reported to be both positive and negative [6] and can be the source of a few issues, including waste transport or illegal waste dumping. In response to a question related to this cost disparity, participants revealed varying opinions (Figure 4.a). About 52.5% of participants believed that different levy rates are problematic, and consistency is needed to avoid unintended consequences and unnecessarily complications. However, about 32.5% of participants indicated that differences in levy rates between metro and regional areas are practical and effective; they even stated that further discount and waivers are needed for regional areas (Figure 4. b). Hence, further investigation is needed to fully understand how discrepant levy rates can act as a powerful tool for waste management or otherwise.
Cost disparity in levy rate also exists between Australian states. the participants were asked to indicate their opinion about this cost disparity. , The results showed an overwhelming majority of responses (62.5%) supported having a national consistent levy rate (Figure 4. c). About a quarter of responses supported non-uniform levy rates.
Lastly, there is a debate on how to spend the revenue generated from landfill levy imposition. Therefore, this survey targeted respondents’ opinion about the current system of levy revenue distribution. The results, as shown in Figure 4.d, suggested that majority of participants (72.5%) believed that there is a poor distribution of the levy revenue and improvements needed to maximise waste recovery in C&D waste sector. One widely referenced recommendation was to return the revenue to the waste industry. The other interesting finding was that almost one-fifth of participants were unsure about the current distribution system.
|

Reviewer 2 Report
The topic of the paper is interesting, waste treatment is an actual issue in all the world, but the presented paper is poorly presented, the graphs presentation are incosistent, the aims of the paper are confused, and you need to clearify, and the background research needs to improve. More references are needed and the order of references is wrong, need to follow correlative order.
I want to encourage to the authors to imporve the quality of the paper.
Author Response
|
6 |
The graphs presentation are inconsistent |
We have harmonized the graph presentation to make sure they are consistent and understandable to readers; for instance, all figures now have a title. |
|
7 |
The aims of the paper are confused, and you need to clarify |
We clearly outlined the aim and objectives of the study as follows:
Therefore, this study aimed to capture the key stakeholders' perceptions of the landfill levy imposition status quo in Australia, including its effectiveness in reducing C&D waste landfilling. Two specific objectives were
1. To explore C&D waste key stakeholders perceptions of the landfill levy 2. To identify the best approach to improve the effectiveness of current landfill levy regimes
|
|
8 |
The background research needs to improve |
We improved the background research in the introduction. |
|
9 |
More references are needed and the order of references is wrong, need to follow correlative order. |
We added more references to the introduction and discussion. Also we fixed the issue with the order of references. |

Reviewer 3 Report
The papaer is inetersting and easy to read, it provides sufficient information on the stakeholders opinion and lead to clear improvement recommendations. I would sat that authors made the tangible value out of only six survey questions. In my view some improvements are needed in the survey results presentation. On the pages from 10 to 14 several figures sre presented in the row without titles and comments. I believe it could be technical issue but nevertheless titles and sufficient comments should be attached to each figure. Also the discusiion would be more valuable if generalized opinion of each stakeholder group could be presented to show possibly conflicting ineterests.
Author Response
|
10 |
On the pages from 10 to 14 several figures are presented in the row without titles and comments. I believe it could be technical issue but nevertheless titles and sufficient comments should be attached to each figure. |
In addition to harmonizing the format of all figures, we added titles to all figures, and fore each figure there is a result statement in the text. For instance for Figure 4 which contain multiple graphs the result presentation is as follows:
In some Australian states, there is a discrepancy in the landfill levy rate between regional and metropolitan areas. In response to a question related to this cost disparity, participants revealed varying opinions (Figure 4.a). About 52.5% of participants believed that different levy rates are problematic, and consistency is needed to avoid unintended consequences and unnecessarily complications. However, about 32.5% of participants indicated that differences in levy rates between metro and regional areas are practical and effective; they even stated that further discount and waivers are needed for regional areas (Figure 4. b). Hence, further investigation is needed to fully understand how discrepant levy rates can act as a powerful tool for waste management or otherwise. In terms of consistency in levy rate across Australia, an overwhelming majority of responses (62.5%) supported having a national consistent levy rate (Figure 4. c). The way that the levy revenue is to be distributed has always been a point of argument. Therefore, this survey targeted respondents’ opinion about the current system of levy revenue distribution. The results, as shown in Figure 4. d, suggested that majority of participants (72.5%) believed that there is a poor distribution of the levy revenue and improvements needed to maximise waste recovery in C&D waste sector. One widely referenced recommendation was to return the revenue to the waste industry.
Furthermore, we separate the two graphs previously presented under Figure 4. |
|
11 |
Also, the discussion would be more valuable if generalized opinion of each stakeholder group could be presented to show possibly conflicting interests. |
While we appreciate and value the point of this comment, we firmly believe that data analysis based on disproportionate representation of stakeholder groups can produce misleading information |

Round 2
Reviewer 1 Report
Methods can be improved.
Conclusions must be improved including some critical considerations about the results of the study.
Author Response
- Methods can be improved:
We restructured and improved methodology section. As a result, the methodology is divided into three subheadings:
- Sample and data collection
- Survey design
- Data analysis and presentation
2. Conclusions must be improved including some critical considerations about the results of the study
We improved the conclusion part to better represent the implication of research findings. We are happy to take on board further comments and suggestions if the changes made are not sufficient.
Reviewer 2 Report
The idea of ​​the article is good. There is a clear interest in recycling and waste management that society makes of the waste that is generated. The article has improved but still has inconsistencies and aspects to improve, among others: 1. page 2, line 77, talks about landfill levy generated, in what area? Australia, world? 2. Page 3, line 129, talks about new China’s waste policy, can you expand more information on this? 3. The graphics need to be improved, the text on the Y axis does not look good 4. There are some references where information is missing such as in 4: Missing in the name of the publication: Journal, Conference,…. publication 5. References remain uncorrelated on paper 6. In the conclusions, I would like to define the type of taxes that the authors believe would be effective to implement, according to their perception / research 7. The conclusions talk about 4 characteristics landfill levy in Australia, (Line 402 in conclusions) including the effectiveness of current Austalian landfill levy regimes, can you define them more accurately and broadly?Author Response
- page 2, line 77, talks about landfill levy generated, in what area? Australia, world?
The location is Australia which we added it to the specified line.
2. Page 3, line 129, talks about new China’s waste policy, can you expand more information on this?
That’s a very valid comment, thanks! To address this comment, we added the following paragraph to the introduction
To add to the complexity, recent anti-waste movements by foreign countries such as China, Thailand, Philippines and Malaysia that ban the import of recyclable from developed nations [4, 5] has left Australia with alarming rates of waste stockpiling [6, 7]. This is mainly because the waste producers can no longer avoid landfill levies or recovery operation fees by shipping waste overseas. Although this policy only focuses on certain types of metals, textiles, plastic, cardboards used in packaging and not all C&D waste, the announced level of acceptable contamination is a real hurdle to the export of C&D waste from Australia. Some Australian organisations have claimed that the ban diminishes the ability of material recovery facility (MRF) operators to market sorted recyclables, and consequently stockpiling, and more landfilling will likely to occur [8].
3. The graphics need to be improved, the text on the Y axis does not look good
We fixed the issue with the Y axis text, we are happy to make further improvements, it the reviewer could kindly direct us.
4. There are some references where information is missing such as in 4: Missing in the name of the publication: Journal, Conference,. publication
We checked all references in the reference list and fixed the issues with incomplete references
5. References remain uncorrelated on paper
We checked the manuscript and can confirm that the references are in correlated order. The only source of confusion might be related to the references that are for the first time used in the introduction and then for the second time in the discussion (for instance) which changes its correlative order. Let’s say we used (Shooshtarian et al, 2019) [1] in the introduction but also in the discussion with some other references like [1, 22,23} the final product is not in correlative order. Please let us know if the misunderstood this comment.
6. In the conclusions, I would like to define the type of taxes that the authors believe would be effective to implement, according to their perception / research
Our understanding is that there is no variation in types of landfill taxes in Australia, the only difference is about its rates differently imposed in Australian jurisdictions (or between regional and metro areas); we explained the differences in results and discussion. Please let us know if we misunderstood the comment.
7. The conclusions talk about 4 characteristics landfill levy in Australia, (Line 402 in conclusions) including the effectiveness of current Austalian landfill levy regimes, can you define them more accurately and broadly?
Their definitions are presented in results section, also they were further discussed in discussion section.